# Multiplex Analysis to Unravel the Mode of Antifungal Activity of the Plant Defensin HsAFP1 in Single Yeast Cells

**DOI:** 10.3390/ijms23031515

**Published:** 2022-01-28

**Authors:** Caroline Struyfs, Jolien Breukers, Dragana Spasic, Jeroen Lammertyn, Bruno P. A. Cammue, Karin Thevissen

**Affiliations:** 1Centre of Microbial and Plant Genetics, Department of Microbial and Molecular Systems (M^2^S), KU Leuven, Kasteelpark Arenberg 20, 3001 Leuven, Belgium; caroline.struyfs@kuleuven.be (C.S.); bruno.cammue@kuleuven.be (B.P.A.C.); 2Biosensors Group, Department of Biosystems (BIOSYST), KU Leuven, Willem de Croylaan 42, 3001 Leuven, Belgium; jolien.breukers@kuleuven.be (J.B.); dragana.spasic@kuleuven.be (D.S.); jeroen.lammertyn@kuleuven.be (J.L.)

**Keywords:** continuous microfluidics, OSTE+ microwell array, yeast, single cell, HsAFP1

## Abstract

Single cell analyses have gained increasing interest over bulk approaches because of considerable cell-to-cell variability within isogenic populations. Herein, flow cytometry remains golden standard due to its high-throughput efficiency and versatility, although it does not allow to investigate the interdependency of cellular events over time. Starting from our microfluidic platform that enables to trap and retain individual cells on a fixed location over time, here, we focused on unraveling kinetic responses of single *Saccharomyces cerevisiae* yeast cells upon treatment with the antifungal plant defensin HsAFP1. We monitored the time between production of reactive oxygen species (ROS) and membrane permeabilization (MP) in single yeast cells for different HsAFP1 doses using two fluorescent dyes with non-overlapping spectra. Within a time frame of 2 min, only <0.3% cells displayed time between the induction of ROS and MP. Reducing the time frame to 30 s did not result in increased numbers of cells with time between these events, pointing to ROS and MP induction as highly dynamic and correlated processes. In conclusion, using an in-house developed continuous microfluidic platform, we investigated the mode of action of HsAFP1 at single cell level, thereby uncovering the close interdependency between ROS induction and MP in yeast.

## 1. Introduction

Individual cells within clonal populations grown in static environments can show significant cell-to-cell variability, also denoted as phenotypic or non-genetic heterogeneity. This is reflected, amongst others, in their gene expression, protein content, morphology and responses upon external stress or drug treatment [1,2,3,4]. For example, yeast cells exhibit cell-to-cell heterogeneity upon multiple stress responses, e.g., copper resistance and heat shock resistance are both correlated with replicative cell age [1,5]. Non-genetic heterogeneity in yeast cells is also observed during the real-time monitoring of single cell responses upon antifungal treatment. Indeed, different yeast subpopulations were, for example, detected with respect to amphotericin B (AmB)-induced superoxide radical production, based on their intracellular levels and timing [6,7]. This non-genetic heterogeneity can give rise to non-responsive subpopulations upon treatment, which can cause treatment failure and the recurrence of infections, pointing out the importance of single cell analyses.

Nowadays, multiple techniques enable to study single cell variability, such as flow cytometry, microscopy, microfluidics and single-cell sequencing. Hereof, flow cytometry is considered as a standard single cell technique due to its high-throughput and the ability to analyze multiple parameters simultaneously [8]. However, it is also associated with a number of drawbacks, as it does not allow to monitor cellular growth and interactions, secreted cellular products or kinetic responses [7,9,10,11]. Therefore, we previously optimized a poly(ethylene glycol) (PEG)-grafted off-stoichiometry thiol-ene epoxy (OSTE+) microwell array, integrated with a polydimethylsiloxane (PDMS) microfluidic channel that enables automated and effective single yeast cell seeding and washing of non-seeded yeast cells [12]. The microwells assure confinement of single cells over time, thus allowing spatiotemporal resolution.

In this study, we focused on unraveling kinetic responses of single yeast cells upon treatment with the plant defensin HsAFP1 from coral bells (*Heuchera sanguinea*), which displays broad-spectrum antifungal activity (MIC of 25 µg/mL) [13,14,15]. Moreover, Cools and colleagues demonstrated that HsAFP1 is not cytotoxic to human liver tumor cells (HepG2) [15]. HsAFP1 is known to be internalized in *Saccharomyces cerevisiae* cells, which is at least partially endocytosis-mediated and depends on the proper functioning of Bst1p. Bst1p is an inositol deacylase that removes the acyl-chain from glycosylphosphatidylinositol (GPI)-anchors, pointing to the importance of GPI-anchor remodeling enzymes in HsAFP1’s internalization [16,17]. Subsequently, cell membranes are permeabilized, resulting in approximately 25% and 85% of cells with compromised membranes upon treatment with low (25 µg/mL) and high (285 µg/mL) HsAFP1 concentrations, respectively, after 150 min at 30 °C [16]. As only a small subpopulation (<1%) of HsAFP1-treated cells display internalization of HsAFP1 while membranes are still intact, a three-step killing process of HsAFP1 in *S. cerevisiae* was previously proposed: HsAFP1 accumulates at the cell surface and internalizes, immediately followed by membrane permeabilization [16]. In addition, HsAFP1 induces ROS production and regulated cell death in yeast cells [18]. As the kinetic responses and interdependency of these cellular events upon HsAFP1 treatment were not yet investigated, we focused on ROS induction and membrane permeabilization. By investigating the interplay between both cellular events induced upon HsAFP1 treatment, we observed that these two events are highly dynamic and tightly correlated.

## 2. Results

### 2.1. Validation of the Continuous Microfluidic Platform and of Multiplexing Fluorescent Dyes

To investigate the responses of single yeast cells with spatiotemporal resolution, we previously established a continuous microfluidic platform with a PEG-grafted OSTE+ microwell array [12]. Using that platform, single yeast cells were successfully confined in microcavities and monitored for 4 h without affected single yeast cell viability [12]. In this work, we used this microfluidic platform to investigate the interdependency of cellular events in single yeast cells over time upon treatment with the antifungal plant defensin HsAFP1. Our platform was compared to flow cytometry based on the percentage of propidium iodide (PI)-positive yeast cells, as PI can only enter cells with compromised membranes and hence serves as a marker for membrane permeabilization. More specifically, yeast cells were treated with different concentrations of HsAFP1 (0, 50, 100, 200 and 300 µg/mL) for 1, 2, 3 and 4 h. In both flow cytometry and the microfluidic platform, the medium of the yeast cells was diluted 1/5 to be complient with the microfluidic platform, as this medium dilution is essential for efficient washing of non-seeded cells [12]. Note that HsAFP1 treatment of cells using the microfluidic platform started with a 5 min off-chip pretreatment in bulk, followed by seeding of yeast cells in microwells and washing of non-seeded cells. As this procedure took one hour in total, the on-chip image analysis on the microfluidic platform started from 60 min onwards. The kinetics of cellular events was monitored for a maximum of 3 h on-chip, equaling 4 h of treatment of cells with HsAFP1 (Figure 1). Although the flow rates continuously changed during these seeding and washing steps, the HsAFP1 concentration was kept constant during all steps of the procedure.

The percentages of PI-positive cells obtained using flow cytometry and the continuous microfluidic platform were compared using correlation analyses (Figure 2). The Pearson correlation coefficient ranged from 0.94 to 0.99, pointing to a high degree of correlation of the percentages of PI-positive cells when comparing the two platforms for all tested concentrations at all tested time points. However, after 1 h of HsAFP1 treatment, the correlation between both single cell setups did not match the bisector, thus not pointing to a one-on-one correlation, in contrast to the other tested time points. The high flow rates during cell seeding and washing might cause additional stress for the cells that normalizes when the flow rate decreases after washing of non-seeded cells. This could explain the increased percentage of PI-positive yeast cells when using the microfluidic platform as compared to flow cytometry after 1 h of HsAFP1 treatment. Nevertheless, we previously demonstrated that the platform itself did not affect yeast cell viability [12]. Moreover, the HsAFP1 concentration remained constant during the experiment. HsAFP1 induced membrane permeabilization in a dose-dependent manner with higher HsAFP1 concentrations resulting in higher percentages of PI-positive cells compared to lower HsAFP1 concentrations. For all tested HsAFP1 doses, except 300 µg/mL, the majority of single cells were PI-negative at the start of the image analysis, i.e., after 1 h of treatment (Figure 2A), pointing to the pertinence of evaluating cellular events in single yeast cells over time starting from 1 h HsAFP1 treatment onwards. In conclusion, using our microfluidic platform allowed us to achieve comparable results as the standard flow cytometry technique.

To investigate cellular events in yeast upon HsAFP1 treatment, different fluorescent dyes were previously used: BODIPY to investigate HsAFP1 uptake (BODIPY-labelled HsAFP1), dihydroethidium (DHE) to investigate the induction of reactive oxygen species (ROS) and PI to investigate membrane permeabilization [12,16,17,18]. Notably, DHE is typically utilized for detecting O_2_^−•^ due to its relative specificity for this ROS [19]. Here, we focused on the interdependency of HsAFP1-induced generation of ROS and membrane permeabilization in single yeast cells over time, but due to spectral overlap of DHE (Exc./Em.: 518/606) and PI (Exc./Em.: 535/617), these dyes could not be combined. Therefore, SYTOX Green (SYTOX) (Exc./Em.: 504/523) was tested as an alternative for PI since it can only enter cells with compromised membranes, while it can be combined with DHE [20]. To confirm the interchangeability of PI and SYTOX for evaluating membrane permeabilization, we investigated if similar percentages of PI-positive and SYTOX-positive cells can be observed upon treatment with 100 µg/mL HsAFP1 during 3 h on chip, resulting in 4 h of total HsAFP1 treatment (Figure 3). A minimum of 1008 single yeast cells was monitored for each condition, with imaging being performed every 15 min. No significant differences between the percentage of fluorescently positive cells could be observed, demonstrating the interchangeability of PI and SYTOX for assessing membrane permeabilization of yeast cells.

Next, we tested if using the combination of DHE and SYTOX dyes resulted in similar percentages of fluorescent cells as compared to when using these dyes alone. To do this, similarly to previous experiments, cells were treated with 100 µg/mL HsAFP1 during 3 h on chip, resulting in 4 h of total HsAFP1 treatment. A minimum of 521 single yeast cells was imaged every 15 min for each condition. No significant differences could be observed in the percentages of fluorescent cells when using DHE (Figure 4A) or SYTOX (Figure 4B) alone as compared to using both DHE and SYTOX together (Figure 4). We can thus evaluate ROS induction and membrane permeabilization using the DHE-SYTOX combination on the continuous microfluidic platform without significantly affecting cellular responses.

### 2.2. Interdependency of Cellular Events in Single Yeast Cells upon Treatment with Different HsAFP1 Concentrations

Using the DHE-SYTOX combination of fluorescent dyes, we next assessed ROS generation and membrane permeabilization simultaneously upon treatment of single yeast cells with different doses of HsAFP1 (12.5, 25, 50, 100 and 300 µg/mL) by imaging the arrays every 2 min instead of every 15 min (as in Figure 4) to gain a more detailed understanding on the kinetics. At least 507 single yeast cells were monitored per HsAFP1 dose during 3 h treatment on chip, resulting in 4 h of total HsAFP1 treatment. We first evaluated membrane permeabilization of the single yeast cells. The percentages of cells that were alive (SYTOX-negative) at the start of the image analysis are indicated on the left of Figure 5 and depend on the used HsAFP1 concentration. The percentage of SYTOX-negative cells at the end of the experiments, indicated on the right of Figure 5, decreased with an increasing HsAFP1 dose. These cells comprise both non-responders and apoptotic cells [18]. Previously, the percentage of apoptotic cells in an HsAFP1-treated population (250 µg/mL) was estimated to reach 10% in the undiluted YPH medium after 150 min of treatment [16]. Here, approximately 7% SYTOX-negative cells were observed after 150 min of HsAFP1 treatment (300 µg/mL) in 1/5 diluted YPH medium. So far, it remains unclear which percentage of these SYTOX-negative cells comprises apoptotic cells. HsAFP1 induced cell death in a dose-dependent manner as higher HsAFP1 concentrations resulted in more rapid and higher percentages of membrane permeabilization as compared to lower HsAFP1 concentrations (Figure 5). Significant differences between survival curves were analyzed using the log-rank (Mantel–Cox) test, followed by Bonferroni correction to allow for multiple comparisons. As the total number of comparisons was 10, a *p*-value less than 0.005 was considered statistically significant. Significant differences were found between all treatments (*p* < 0.001), indicating that treatment of yeast cells with different HsAFP1 concentrations affects the number and timing of membrane permeabilization events in a significantly different manner.

Next, the time between ROS induction and membrane permeabilization within a single cell was evaluated (Appendix A). Starting from at least 507 cells for each HsAFP1 concentration, single cells that were already DHE- or SYTOX-positive at the start of the measurements, i.e., at 60 min, and non-responders at the end of the measurements were discarded from the analysis. Consequently, the time points at which cells became DHE-positive and SYTOX-positive were monitored for at least 67 cells for each HsAFP1 concentration. The number of cells that can be retained for the analysis varied greatly with the HsAFP1 concentration: 12.5 (98 cells), 25 (205 cells), 50 (328 cells), 100 (286 cells) and 300 (67 cells) µg/mL. Almost no cells could be observed with time between ROS induction and membrane permeabilization, meaning that they were both DHE-positive and SYTOX-positive from the same time point onwards. Only upon treatment with 50 µg/mL HsAFP1, one cell displayed time between both events. This cell was DHE-positive, while remaining SYTOX-negative within a time frame of 2 min (data not shown). Different HsAFP1 concentrations did not seem to affect the time between ROS induction and membrane permeabilization. From all tested concentrations, we analyzed most single cells upon treatment with 50 µg/mL HsAFP1, as this concentration contained the highest number of single cells that were alive (SYTOX-negative) at the start of image analysis and that were dead (SYTOX-positive) at the end of the experiment (Figure 5). The larger number of analyzed cells could explain why we only observed one cell with time between events upon treatment with 50 µg/mL HsAFP1 when imaging every 2 min and not upon treatment with other HsAFP1 doses.

To rule out the possibility that 2 min between measurements is too long for monitoring the dynamics of these cellular events, we also imaged microwell arrays every 30 s. The cells were treated with 50, 100 and 300 µg/mL during 3 h on-chip, resulting in 4 h of total HsAFP1 treatment. In this setup, no single yeast cells with time between these cellular events could be observed (data not shown). Notably, at least 42 cells were evaluated per HsAFP1 dose. Hence, we can conclude that ROS induction and membrane permeabilization by HsAFP1 are highly correlated and dynamic processes, resulting in a limited amount of single cells (<0.3%) that displayed time between both events. Although only one cell could be observed with time between events, it seems that ROS induction is likely to precede membrane permeabilization.

## 3. Discussion

In this study, we employed an in-house developed continuous microfluidic platform with OSTE+ PEG microwell arrays to investigate single non-adherent cells over time. Standard single cell techniques, such as flow cytometry, often lack spatial and/or temporal resolution, whereas in our setup, single yeast cells are confined in microwells (spatial resolution) and cellular responses are monitored over time (temporal resolution). Our platform was validated by comparison with flow cytometry. Herein, a significant correlation was found between both single cell techniques, indicating that the continuous single cell platform can be employed without causing additional stress to single seeded yeast cells. Moreover, by including five separate microwell arrays in one microfluidic chip as well as automated image acquisition, the throughput and efficiency of the continuous microfluidic platform already significantly increased compared to our previously reported microfluidic design [12].

So far, the antifungal mechanism of action of only a limited number of plant defensins has been described, but for most of these peptides, it does not consist of simple membrane permeabilization [21]. Instead, plant defensins have a more complex mechanism of action that includes activation of signaling pathways and intracellular targets/effectors, such as the production of ROS. Notably, also fungal, insect and vertebrate defensins can interact with specific fungal membrane targets and can induce intracellular mechanisms. More information on the mode of action of these defensins can be found in the following reviews: [21,22,23,24]. Using our microfluidic platform, we investigated the kinetic response of single yeast cells upon treatment with the plant defensin HsAFP1. The mode of action of HsAFP1 was previously studied in bulk, resulting in the induction of ROS and membrane permeabilization [17,18]. Excessive ROS levels damage various molecules, such as nucleic acids, proteins and lipids, which can amount to an increasing membrane permeability, eventually leading to cell death [25]. To our knowledge, the heterogenic response and interdependency of ROS induction and membrane permeabilization upon plant defensin treatment was not yet investigated. In this study, we observed a dose-dependent response in membrane permeabilization upon HsAFP1 treatment, with higher HsAFP1 concentrations resulting in more rapid and increased number of membrane permeabilizing events compared to low HsAFP1 concentrations. In addition to those, a population of SYTOX-negative cells (i.e., without permeabilized membranes) was also observed at the end of the experiment, which includes both non-responders, which can become SYTOX-positive at a later time point or remain long term non-responders, and apoptotic cells. Previously, the percentage of apoptotic cells in an HsAFP1-treated population (250 µg/mL) was estimated to reach 10% in the undiluted YPH medium after 150 min of treatment [16,18]. Here, approximately 7% SYTOX-negative cells were observed after 150 min of HsAFP1 treatment (300 µg/mL) in 1/5 diluted YPH medium. So far, it remains unclear which percentage of these SYTOX-negative cells comprises apoptotic cells.

Single yeast cells also displayed a heterogeneous response upon treatment with HsAFP1: at every evaluated time point, certain single cells displayed membrane permeabilization, whereas others did not, and this ratio varied over time. This is in line with a previous study by Kumar and colleagues, using a similar continuous microfluidic platform to study the heterogenous response of single yeast cells after treatment with the antifungal drug AmB [6]. They monitored either the induction of ROS or membrane permeabilization in independently treated yeast populations. However, here, by multiplexing different fluorescent dyes, we could for the first time simultaneously monitor ROS induction and membrane permeabilization at single cell resolution, thereby evaluating the time between those two events. Within a time frame of 2 min, one cell could be observed with time between events. Reducing the time frame to 30 s did not result in increased numbers of single cells with time between events, pointing to highly dynamic and correlated processes. Although only one cell could be observed with time between events, it seems that ROS induction precedes membrane permeabilization, but the interval between both events is very limited.

As a next step, it would be interesting to evaluate whether the absence of time between ROS induction and membrane permeabilization is a general feature of ROS-inducing antifungals (such as AmB and miconazole) or if other antifungals would demonstrate a more prolonged interval between both events. Moreover, in the past, the internalization of HsAFP1 has been assessed by labeling the peptide with a green fluorescent dye, i.e., BODIPY [16,17]. In the future, it would therefore be interesting to evaluate peptide uptake, ROS induction and membrane permeabilization simultaneously by using non-overlapping fluorescent dyes, such as BODIPY, CellROX Deep Red and PI, respectively [16,17,26]. Lastly, here, we kept the concentration of HsAFP1 constant throughout the experiment. However, in plant–phytopathogen systems, a local and transient increase in plant defensin concentration is most likely to occur, whereas when treating a fungal infection with intravenous administration of a defensin-based drug, the concentration of the active substance will decrease over time. Therefore, in follow-up experiments, it would be very interesting to assess the effect of varying the peptide concentration over time via the automated reagent delivery on chip.

## 4. Materials and Methods

### 4.1. Materials and Reagents

#### 4.1.1. Materials for Microfluidic Chip Fabrication

Similar to our previous work [12,27], stamp-molding materials for OSTE+ microwell arrays (i.e., UV initiator 1-hydroxycyclohexyl phenyl ketone, poly(ethylene glycol) methacrylate with an average molecular weight of 500 and poly(ethylene glycol) methyl ether methacrylate with an average molecular weight of 2000) were acquired from Sigma-Aldrich. Borosilicate glass microscope slides (1 mm thickness) were supplied by Carl Roth. Dow Corning and BAP Medical B.V. provided PDMS Sylgard 184 silicone elastomer kit and biopsy punches of 1 mm diameter, respectively.

#### 4.1.2. Strains and Materials for Culturing and Staining Yeast Cells

*Saccharomyces cerevisiae* BY4741 cells (Invitrogen, Waltham, MA, USA) were cultured in the following liquid media: YPD (yeast extract (10 g/L; LabM, Heywood, UK), peptone (20 g/L; LabM) and glucose (20 g/L; Sigma-Aldrich, St. Louis, MO, USA)) and yeast cell medium PDB/YPD (potato dextrose broth (19.2 g/L; BD), yeast extract (2 g/L), peptone (4 g/L) and glucose (4 g/L)) adjusted to pH 7 with 50 mM HEPES (Sigma-Aldrich) and diluted 1/5 using distilled water (dH_2_O), which is required for the efficient washing of non-seeded cells. The plant defensin HsAFP1 was recombinantly produced, as described previously [28]. PI was purchased from Sigma-Aldrich, and DHE and SYTOX staining were obtained from ThermoFisher Scientific (Waltham, MA, USA).

### 4.2. Fabrication of Microfluidic Chips with Microwell Arrays

The fabrication of OSTE+ PEG 500/2000 arrays with 57,600 microwells was previously described [12,27]. In short, a silicon microwell array mold was fabricated by deep reactive ion etching, a PDMS micropillar stamp was prepared by PDMS casting, and borosilicate microscope glass slides were treated with methacrylate silane. Next, the OSTE+ solution was prepared by mixing compounds A and B from OSTEMER 322 Crystal Clear in a mass ratio of 1.25 to 1. The mixture was vortexed for 2 min and degassed for 15 min. A few drops of OSTE+ were spin-coated on the silanized glass slide at 2500 rpm for 30 s, after which the micropillar stamp was pushed into OSTE+. Then, the polymer was exposed to UV (24 mW/cm^2^) for 2 min, after which the PDMS stamp was peeled off and the dimensions of the microwells were verified via a 3D optical profilometer (S lynx, Sensofar Metrology, Terrassa, Spain). An equimolar solution of PEG 500 and PEG 2000, provided as 50 w/w% in water, consisted of 6.25 w/w% PEG 500, 25 w/w% PEG 2000, 25 w/w% water, and 1% UV initiator in ethanol. The OSTE+ microwell array was grafted with PEG 500/2000 after the first UV cure by submerging the slide in the PEG solution. The array was then exposed to UV (24 mW/cm^2^) during 5 min, after which the slides were rinsed with dH_2_O and blow-dried with a nitrogen gun. Here, five microwell arrays were positioned on one microfluidic chip to increase the experimental throughput. The microfluidic channel was fabricated with PDMS in a 10 to 1 ratio of the base and curing agent, and then mixed, degassed and poured into a 3D-printed mold with 200 μm high channels. After a thermal cure at 60 °C for 3 h, PDMS was peeled off from the mold and cut into properly sized pieces. Inlet and outlet holes (1 mm) were punctured using a biopsy punch. The PDMS surface was activated with oxygen plasma (20 sccm, 200 mTorr, and 30 W for 45 s), placed on top of the slide with the microwell array and placed in an oven at 60 °C overnight.

### 4.3. Preparation of Yeast Cells

Exponentially growing *S. cerevisiae* BY4741 cells in YPD (grown for 5 h at 30 °C and 250 rpm starting from OD_600 nm_ = 0.3) were pelleted and washed. The pellet was resuspended in PDB/YPD with 50 mM HEPES pH 7 diluted 1/5 using dH_2_O (OD_600 nm_ = 1; equaling 10 million yeast cells/mL). Induction of ROS species and membrane permeabilization were evaluated by labeling cells with DHE staining (17 µM) and PI (3 µM) or SYTOX (1 µM), respectively.

### 4.4. Membrane Permeabilization Evaluated by Flow Cytometry

Yeast cells were incubated with HsAFP1 or Milli-Q (control) at room temperature in bulk, followed by flow cytometry on a BD Influx cell sorter. A total of 100,000 cells were monitored for fluorescence at 610/20 nm (FL11_λ_ex_ = 561 nm) to detect membrane permeabilization, using PI.

### 4.5. ROS Induction and Membrane Permeabilization Evaluated by the Continuous Microfluidic Platform

Yeast cells were pre-incubated with HsAFP1 or Milli-Q (control) at room temperature during 5 min in bulk, after which cell seeding in microwell arrays started. The chip was placed under the microscope, and the outlet was connected to a Nemesys syringe pump (Cetoni, Korbußen, Germany). A pipet tip filled with cell suspension was inserted in the inlet, and the cells were pulled through the microfluidic chip at 5 µL/min during 30 min to allow single yeast cell seeding. To remove unseeded cells, yeast cell medium was flowed during 15 min at 10 µL/min. During seeding and washing steps, the HsAFP1 concentration remained constant. Yeast cells were monitored under an inverted microscope (Nikon Ti-Eclipse, Tokyo, Japan) using a 40 × objective (CFI Plan Apochromatic, numerical aperture (NA) 0.55, working distance (WD) 2.1 mm; Nikon). Imaging of the cells was done using bright field, FITC and TRITC fluorescence microscopy every 15 min, 2 min or 30 s using automated image acquisition during a period of 3 h on-chip, equaling 4 h of HsAFP1 treatment in total. A minimum of 42 single yeast cells were evaluated per condition.

### 4.6. Image Processing and Statistical Analysis

Fluorescence microscopy images were visualized using ImageJ 1.52 p (National institutes of Health). During image analysis, only single-cell microwells were taken into account, whereas all microwells containing none or multiple cells were discarded. Image analysis was done manually. The percentage of DHE-, PI- or SYTOX-positive cells was determined via visual assessment of the time stacks of FITC- or TRITC-filter images and corresponding bright field images. When determining the time between cellular events, single cells that were already DHE- or SYTOX-positive at the start of the measurements, i.e., at 60 min, and non-responders at the end of the measurements were discarded. Herein, the time points at which cells became DHE-positive and SYTOX-positive were monitored, and the time between these events per cell was subsequently determined. Statistical analyses were performed with GraphPad Prism (GraphPad v5.03 Software, San Diego, CA, USA). Pearson correlation analyses were used to determine the correlation between the percentage of PI-positive cells obtained with flow cytometry versus the continuous microfluidic platform at different time points and HsAFP1 doses. To test cell survival upon the use of different fluorescent dyes, survival analyses were performed using the likelihood ratio test on the Cox proportional hazards regression for survival data (α = 0.05), taking into account multiple biological replications. When comparing multiple survival curves to evaluate if treatment conditions affect cell survival, significant differences were determined using the log-rank (Mantel–Cox) test with Bonferroni correction to allow for multiple comparisons. Herein, *p* < the Bonferroni-corrected threshold (0.05 divided by K) was considered statistically significant, in which K is defined as the total number of comparisons. For all other experiments, *p* < 0.05 was considered statistically significant. All data were visualized using GraphPad Prism.

## 5. Conclusions

In this work, we used an in-house developed continuous microfluidic platform to monitor single cells with spatiotemporal resolution. More specifically, we investigated cellular responses, and the interdependency thereof, in single *S. cerevisiae* cells induced by the plant defensin HsAFP1. Experimental results demonstrated that our optimized single cell platform resulted in a similar outcome as the standard flow cytometry technique. By employing two fluorescent dyes with non-overlapping spectra (i.e., DHE and SYTOX), the interdependency of cellular events, being ROS induction and membrane permeabilization, could be investigated. Nevertheless, within a time frame of 2 min, almost no single cells could be observed with time between these events, pointing to highly dynamic and correlated processes. In summary, the optimized continuous microfluidic platform with OSTE+ PEG microwells can be used to investigate a myriad of cellular processes in different cell types to gain a more profound insight at a higher resolution.

## Figures and Tables

**Figure 1 ijms-23-01515-f001:**
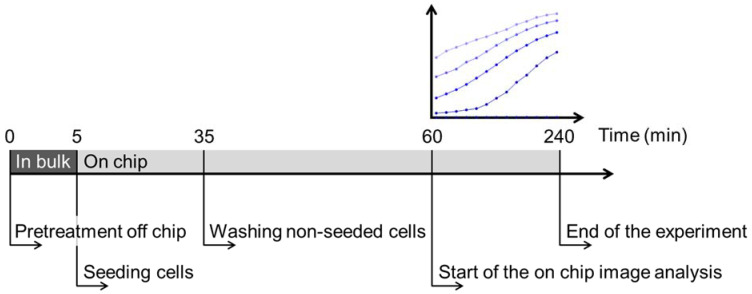
Schematic overview of the timing of single cell experiments using the optimized continuous microfluidic platform with a PEG-grafted OSTE+ microwell array. Cells are pretreated in bulk with HsAFP1 during 5 min, after which cells are seeded on-chip (30 min). After the seeding step, non-seeded cells are washed (15 min), followed by the start of the on-chip image analysis from 60 min onwards. The kinetics of cellular events is monitored during 3 h on-chip, equaling 4 h of treatment of cells with HsAFP1.

**Figure 2 ijms-23-01515-f002:**
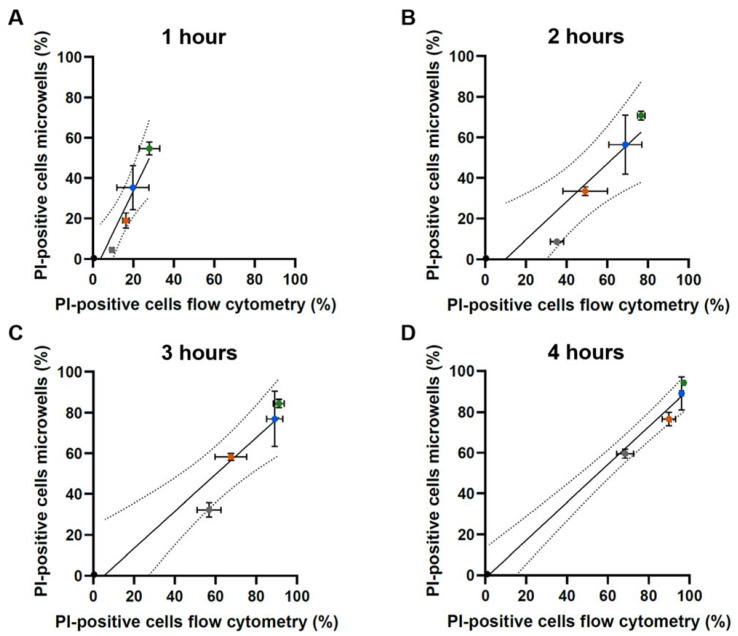
Pearson correlation analysis between the percentages of single yeast cells with compromised membranes obtained by flow cytometry (x-axis) and by the continuous microfluidic setup (y-axis). Yeast cells were treated with 0 (black), 50 (gray), 100 (orange), 200 (blue) and 300 (green) µg/mL HsAFP1 during 1 h (**A**), 2 h (**B**), 3 h (**C**) and 4 h (**D**) at room temperature using PI that allows to evaluate membrane permeabilization. Note that image analysis on-chip can only start from 1 h treatment of cells with HsAFP1 onwards. Means, standard deviations (SD) of three independent biological experiments (*n* = 3) and 95% confidence intervals of the Pearson correlation (dashed lines) are plotted, and Pearson r coefficients and *p*-values are indicated.

**Figure 3 ijms-23-01515-f003:**
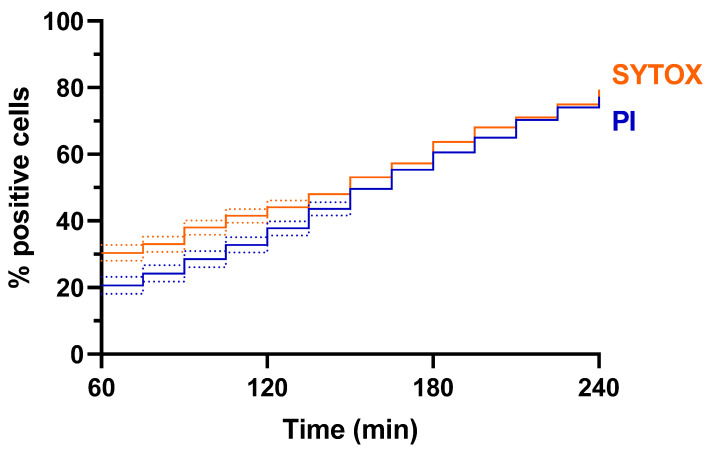
Comparing the percentages of fluorescently positive single yeast cells upon treatment with HsAFP1 evaluated by PI or SYTOX. Yeast cells were treated with 100 µg/mL HsAFP1 during 3 h on-chip, resulting in 4 h of total treatment at room temperature using PI (blue) or SYTOX (orange) for evaluating membrane permeabilization. Images were taken every 15 min. A minimum of 1008 cells was evaluated for every condition. Significant differences were determined using a likelihood ratio test on the Cox proportional hazards regression for survival data (α = 0.05). Dotted lines represent asymmetrical 95% confidence intervals. No significant differences could be observed.

**Figure 4 ijms-23-01515-f004:**
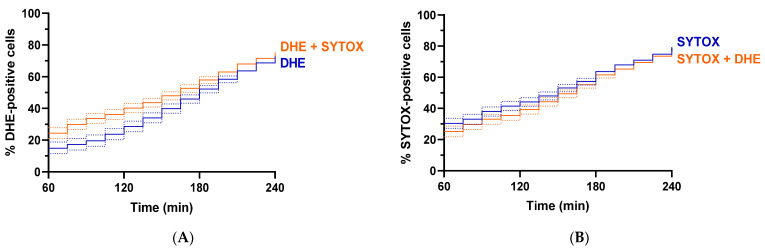
Comparison of % fluorescently positive single yeast cells upon treatment with HsAFP1 evaluated by DHE and SYTOX. Yeast cells were treated with 100 µg/mL HsAFP1 during 3 h on chip, resulting in 4 h of total treatment at room temperature using DHE and/or SYTOX to evaluate ROS induction or membrane permeabilization. Images were taken every 15 min. The percentages of fluorescent cells when using DHE (**A**) or SYTOX (**B**) alone (blue) as compared to using both DHE and SYTOX together (orange). A minimum of 521 cells was evaluated for every independent biological experiment; 3 independent biological experiments were included (*n* = 3). Significant differences were determined using a likelihood ratio test on the Cox proportional hazards regression for survival data (α = 0.05). Dotted lines represent asymmetrical 95% confidence intervals. No significant differences could be observed.

**Figure 5 ijms-23-01515-f005:**
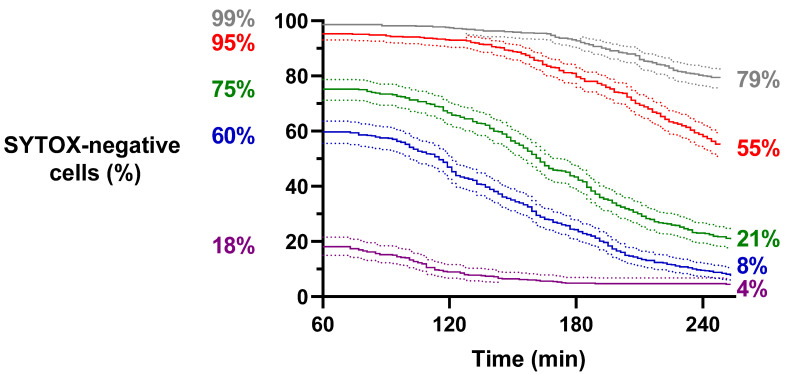
Survival analysis of single yeast cells upon treatment with HsAFP1 evaluated by SYTOX. Yeast cells were treated with 12.5 (grey), 25 (red), 50 (green), 100 (blue) and 300 (purple) µg/mL HsAFP1 during 3 h treatment on-chip, resulting in 4 h of total treatment at room temperature using DHE and SYTOX to evaluate membrane permeabilization. Images were taken every 2 min. The percentages of SYTOX-negative cells at the start and the end of image analysis are indicated per condition on the left and right, respectively. A minimum of 507 cells was evaluated for every HsAFP1 concentration, originating from 2 independent biological experiments (*n* = 2). The log-rank (Mantel–Cox) test, followed by Bonferroni correction to allow for multiple comparisons, was used to evaluate the differences between curves. Dotted lines represent asymmetrical 95% confidence intervals. All treatments resulted in significantly different percentages of SYTOX-negative cells over time.

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
