# Peer review of "Multiplex Analysis to Unravel the Mode of Antifungal Activity of the Plant Defensin HsAFP1 in Single Yeast Cells"

_ijms, 2022, doi:10.3390/ijms23031515_

Round 1

Reviewer 1 Report

The manuscript proposed by Caroline Struyfs colleagues is proposing new data on the mechanisms of action of an antifungal plant defensin HsAFP1 on single yeast Saccharomyces cerevisiae. To obtained those results the authors used a cytometry based method through an in-house developed continuous microfluidic platform and two dies with non-overlapping spectra (namely DHE and SYTOX) to follow ROS Production and membrane permeabilization. They performed a comparison between their microfluidic platform and a more conventional flow cytometry approach and are concluding that their in-house developed continuous microfluidic platform with OSTE+PEG microwells represents a pertinent method to monitoring cellular process at least on their yeast model. This is a rather elegant tool for investigating cellular processes.

The manuscript is well written with an exhaustive up-to-date bibliography to support the reader in its understanding on the mechanisms of action of the antifungal plant defensins. They are few adjustments that are recommended within the reference section and within the rest of the manuscript.

Reference section a critical reading is recommended for a better homogeneity. Ref 1, is is Plos Biol. or as stated Plos Biology?; Ref 6 Lab. Chip. And not Lab Chip; Ref 10, Biochim in italic; Ref 11, may be the year should be in Bold. Ref 22, Biochem. Biophys. Res. Commun.

Manuscript itself, minor adjustments are recommended prior final acceptance or edition.

In section 1 Introduction: line 42 add a between “as” and “standard”

In section 2 Results: Line 76, is firstly needed?; Is “In both setups” related to the in-house continuous microfluidic platform and to flow cytometry? This is not clear; Figure 1 is mentioning a time scale with 60 min, in the capture of this figure it is mentioned a 15 min wash. If this is following the time point at 35 minutes we should expect to find 50 min rather than 60. Or I am misunderstanding the procedure.

In section 4 Materials and Methods: Line 309, as first appearance Saccharomyces rather than S. same comment in section 1 Introduction line 54; line313 what is dH2O? This should be different from MilliQ water mentioned line 348.

Section 5 Conclusion: line 397 and 398, remove “recultered in a similar outcome”. Already mentioned within the same line 397.

Some additional comments

What is the opinion of the authors on any limitation associated of the size of the cells to investigate? A yeast culture is rather “clonal” and will provide cells with the same size. Would their approach usable on a mixture of cell types and is there any limitation due to cells that are developing within a culture in a non-synchronized process of development? Would such a platform applicable on mucoid cells?

The authors are mentioning within lines 51 to 53 the properties of the HsAFP1 antifungal peptite on fungi. However, they are not reporting in the proposed manuscript the MIC value regarding Saccharomyces. The authors are using different concentrations of their peptide of interest (from 50 to 300 ug/mL) without mentioning what is the MIC value on Saccharomyces? Where is positioning the MIC value within the concentration range tested? Another comment is related to the kinetic of killing of Saccharomyces when in contact with HsAFP1? Such information should be provided to the reader.

It would have been also interesting for the reader to have in the discussion sections some information on the mechanisms of action of non-plant antifungal defensins.

Why in some experiments the authors recorded data from 1000 cells and in other experiments on 500 cells?

Why not combining Fig 4A and Fig4B to have only one figure 4?

Figure 5, the percentages of STYTOX-negative cells at the end of the experiment (right values) could be aligned to the corresponding curves.

The experimental design is adapted and the results obtained significant and robust enough for publication in IMS after minor adjustments.

Author Response

Response to Reviewer 1

Point 1: Reference section a critical reading is recommended for a better homogeneity. Ref 1, is Plos Biol. or as stated Plos Biology?; Ref 6 Lab. Chip. And not Lab Chip; Ref 10, Biochim in italic; Ref 11, may be the year should be in Bold. Ref 22, Biochem. Biophys. Res. Commun.

Response 1: We agree with the reviewer’s comment and modified the reference section (lines 431-482).

Point 2: Line 42 add a between “as” and “standard”

Response 2: We included “a” between “as” and “standard” (line 42).

Point 3: Line 76, is firstly needed?

Response 3: We removed the term “firstly” (line 80).

Point 4: Is “In both setups” related to the in-house continuous microfluidic platform and to flow cytometry? This is not clear.

Response 4: We rephrased this sentence to assure that it is clear for the reader (lines 84-85).

Point 5: Figure 1 is mentioning a time scale with 60 min, in the capture of this figure it is mentioned a 15 min wash. If this is following the time point at 35 minutes we should expect to find 50 min rather than 60. Or I am misunderstanding the procedure.

Response 5: Indeed, the reviewer correctly observed that the preparatory phase would be – in theory - completed in approximately 50 min after the start of the experiment. However, in between steps, time is also lost because of the practical manipulations, such as moving yeast from the Eppendorf tube (during pretreatment) to the microwell array. In addition, before the image analysis can start, the image acquisition software is prepared and bright field images are taken. Taken together, this results in a feasible start of the fluorescence image analysis only upon 60 min after starting the treatment.

Point 6: Line 309, as first appearance Saccharomyces rather than S. same comment in section 1 Introduction line 54;

Response 6: We modified S. to Saccharomyces in the Introduction section (line 55) and Materials and Method section (line 321).

Point 7: Line 313, what is dH2O? This should be different from MilliQ water mentioned line 348.

Response 7: We included the meaning of dH2O (line 325).

Point 8: Line 397 and 398, remove “resulted in a similar outcome”. Already mentioned within the same line 397.

Response 8: We removed “resulted in a similar outcome” (line 411).

Point 9: What is the opinion of the authors on any limitation associated of the size of the cells to investigate? A yeast culture is rather “clonal” and will provide cells with the same size. Would their approach usable on a mixture of cell types and is there any limitation due to cells that are developing within a culture in a non-synchronized process of development? Would such a platform applicable on mucoid cells?

Response 9: When optimizing the continuous microfluidic platform, we investigated both yeast cells and human B cells (Breukers et al. 2021). We demonstrated that we can efficiently seed both cell types using microwell arrays with different diameter sizes. If considerable size differences within a cellular culture would occur, then the diameter of the microwells can be based on the largest cells in order to seed all cells, although this will result in an increased chance of capturing more than 1 cell (with smaller size) in one microwell. In that case, the concentration of the cell suspension can be reduced, so that chances of seeding multiple cells within one microwell are reduced or microwells in which multiple cells reside, can be discarded during the image analysis. Alternatively, triangular microwells were used in the past for seeding single cells that were significantly smaller as compared to the microwell size to allow cellular growth (Park et al. 2010). This strategy could also be used to investigate cellular cultures with a wide range in cell size, as long as the cellular suspension is not too concentrated. Implementing microwells with different size or shapes is very straightforward using the polyethylene glycol off-stoichiometry thiol-ene-epoxy microwell array we previously optimized (Breukers et al. 2021). To date, we focused on non-adherent cells, but also seeding of adherent cells could be tested using our microfluidic platform, although the washing of non-seeding cells should be evaluated, as it is undesirable that these cells adhere to the top surface of the microwell array.

Point 10: The authors are mentioning within lines 51 to 53 the properties of the HsAFP1 antifungal peptide on fungi. However, they are not reporting in the proposed manuscript the MIC value regarding Saccharomyces. The authors are using different concentrations of their peptide of interest (from 50 to 300 µg/mL) without mentioning what is the MIC value on Saccharomyces? Where is positioning the MIC value within the concentration range tested? Another comment is related to the kinetic of killing of Saccharomyces when in contact with HsAFP1? Such information should be provided to the reader.

Response 10: We agree with the reviewer’s comment and included the MIC value of HsAFP1, being 25 µg/mL on Saccharomyces cerevisiae in the Introduction section (line 53). The concentration range in Figure 5 includes peptide concentrations both below and above the MIC value: ½ MIC (12.5 µg/mL), MIC (25 µg/mL), 2x MIC (50 µg/mL), 4x MIC (100 µg/mL) and 12x MIC (300 µg/mL). Nevertheless, we would like to point out that we focused here on the fungicidal activity of the plant defensin HsAFP1 and not on its inhibitory action (only the latter is represented via MIC). We selected these peptide concentrations in accordance with our previous publications that focused on the fungicidal activity of HsAFP1, which was observed at HsAFP1 concentrations starting from 25 µg/mL onwards (Cools et al. 2017, Struyfs et al. 2020).

Regarding the kinetics of HsAFP1-induced killing of Saccharomyces, we fully agree with the reviewer’s comment and included the following information in the Introduction section “Subsequently, cell membranes are permeabilized, resulting in approximately 25% and 85% of cells with compromised membranes upon treatment with low (25 µg/mL) and high (285 µg/mL) HsAFP1 concentrations, respectively, after 150 min at 30°C” (lines 59-62).

Point 11: It would have been also interesting for the reader to have in the discussion sections some information on the mechanisms of action of non-plant antifungal defensins.

Point 12: Why in some experiments the authors recorded data from 1000 cells and in other experiments on 500 cells?

Response 12: Our goal was to at least analyze 500 single cells per experiment, in order to draw biologically meaningful conclusions.

Point 13: Why not combining Fig 4A and Fig4B to have only one figure 4?

Response 13: We specifically chose to keep Fig 4A and 4B separate as both graphs contain information on different fluorescent dyes and curves would otherwise be overlapping. To improve the layout of Fig 4, we put Fig 4A and 4B next to each other (line 165).

Point 14: Figure 5, the percentages of SYTOX-negative cells at the end of the experiment (right values) could be aligned to the corresponding curves.

Response 14: We agree with the reviewers comment and aligned the percentages of SYTOX-negative cells at the end of the experiment to the corresponding curves (line 200).

References:

Breukers, J., Horta, S., Struyfs, C., Spasic, D., Feys, H.B., Geukens, N., Thevissen, K., Cammue, B.P.A., Vanhoorelbeke, K., Lammertyn, J. (2021) Tuning the surface interactions between single cells and an OSTE+ microwell array for enhanced single cell manipulation. ACS Applied Materials & Interfaces, 13(2), 2316-2326.

Cools, T.L., Vriens, K., Struyfs, C., Verbandt, S., Ramada, M.H.S., Brand, G.D., Bloch, C., Koch, B., Traven, A., Drijfhout, J.W., Demuyser, L., Kucharíková, S., Van Dijck, P., Spasic, D., Lammertyn, J., Cammue, B.P.A., Thevissen, K. (2017) The antifungal plant defensin HsAFP1 is a phosphatidic acid-interacting peptide inducing membrane permeabilization. Frontier in Microbiology, 2017, 8, 2295.

Park, J.Y., Morgan, M., Sachs, A.N., Samorezov, J., Teller, R., Shen, Y., Pienta, K.J., Takayama, S. (2010) Single cell trapping in larger microwells capable of supporting cell spreading and proliferation. Microfluidics and Nanofluidics, 8(2), 263-268.

Struyfs, C., Cools, T.L., De Cremer, K., Sampaio-Marques, B., Ludovico, P., Wasko, B.M., Kaeberlein, M., Cammue, B.P.A., Thevissen, K. (2020) The antifungal plant defensin HsAFP1 induces autophagy, vacuolar dysfunction and cell cycle impairment in yeast. Biochimica et Biophysica Acta – Biomembranes, 1862:183255.

Reviewer 2 Report

The presented manuscript is quite interesting, first of all, from the point of view of increasing the productivity of testing various natural compounds for antimicrobial activity. It should be noted that the “single cell technology” system used is sufficiently modern and has been periodically used for screening studies in the aspect of discovering for new antibiotics of natural origin for several years. It is interesting that the authors used a two-component model "defensin-yeast cells" in order to assess the time dynamics of the penetration of the studied antimicrobial peptide through the plasma membrane, to identify a quantitative pattern between these parameters and ROS generation, and, as a result, to clarify the membrane-associated mechanism of the molecular action of this peptide. In general, this approach is the future of all screening studies, when it is possible even for extracts or total fractions to detect the presence of the target molecular activity in a large number of natural samples.

It is worth noting the extremely high methodological and experimental component of this study; I have a number of questions and remarks, as follows:

  1. Line 78-79 - the authors took a fairly wide range of effective concentrations, the upper limit of which is 300 μg/ml, which approximately corresponds to 55-60 μM. This is actually quite a high value for plant defensins with antifungal properties. Why was this particular AMP chosen, which, obviously, when compared with homologues, does not have pronounced inhibitory properties against fungi and yeast (MIC more than 10 μM).
  2. Figure 2 is too heavy for objective perception: perhaps, it would be worth rebuilding it by demonstrating the “one concentration/different time” relationship?!
  3. Figure 3 shows data on the accumulation of SYTOX/PI-stained yeast cells only at a concentration of 100 µg/ml. But what about other concentrations in a comparative aspect?!
  4. Fundamentally, the question is extremely important: for what reason did the authors use the approach in which the AMP concentration remained constant throughout the entire observation period? In general, if we try to imagine a “plant-phytopathogen” system, then a situation is unlikely in which the concentration of AMP would remain constant at the site of fungus inoculation, even for up to 4 hours. For defensins as induced polypeptides, the likelihood of a local increase in concentration would be higher. The opposite situation can be assumed, for example, with intravenous administration of a drug containing defensin, when trying to treat a fungal infection - in this case, the concentration of the active substance decreased over time. How is it even possible to extrapolate the current results on S. cerevisiae to a real living system? It would be more correct, at least selectively, to compare the data obtained with a constant, decreasing and increasing concentration of the defensin applied.
  5. Line 215-216 - a surprising result, in which the largest number of SYTOX-positive cells were detected at 25-100 µg/ml. But what about at 300 µg/ml? The cells were completely destroyed?!
  6. For all experiments, there is no data on what was the average number of stained cells in the control, that is, without the addition of the defensin? It is obvious that in any microbial culture there is a natural process of destruction (autolysis).
  7. Is it possible to test these methods on antifungal AMPs, representatives of other structural families, for example, thionins and cyclotides? First of all, in terms of the rate of permeabilization of the fungal cell membrane.
  8. To what extent, from the point of view of the authors, is it possible to compare the membrane-active mechanism of action of AMPs with such well-known antimycotics as amphotericin B and azoles? How is HsAFP1 doing in terms of its cytotoxicity to mammalian cells?

Author Response

Response to Reviewer 2

Point 1: Line 78-79 - the authors took a fairly wide range of effective concentrations, the upper limit of which is 300 μg/ml, which approximately corresponds to 55-60 μM. This is actually quite a high value for plant defensins with antifungal properties. Why was this particular AMP chosen, which, obviously, when compared with homologues, does not have pronounced inhibitory properties against fungi and yeast (MIC more than 10 μM).

Response 1: We thank the reviewer the this interesting remark. The MIC value of HsAFP1, being 25 µg/mL on Saccharomyces cerevisiae is now included in the Introduction section (line 53). The concentration range in Figure 5 includes peptide concentrations both below and above the MIC value: ½ MIC (12.5 µg/mL), MIC (25 µg/mL), 2x MIC (50 µg/mL), 4x MIC (100 µg/mL) and 12x MIC (300 µg/mL). Nevertheless, we would like to point out that we focused here on the fungicidal activity of the plant defensin HsAFP1 and not on its inhibitory action (only the latter is represented via MIC). We selected these peptide concentrations in accordance with our previous publications that focused on the fungicidal activity of HsAFP1, which was observed at HsAFP1 concentrations starting from 25 µg/mL onwards (Cools et al. 2017a, Struyfs et al. 2020).

Point 2: Figure 2 is too heavy for objective perception: perhaps, it would be worth rebuilding it by demonstrating the “one concentration/different time” relationship?!

Response 2: In Figure 2 we aimed at visualizing the correlation between the percentage of PI-positive cells upon treatment with multiple HsAFP1 concentrations when using different experimental setups, being the standard flow cytometry technique and our in-house developed microfluidic platform. Therefore, we choose to plot the percentage of PI-positive cells using flow cytometry (X-axis) and the percentage of PI-positive cells using our microfluidic platform (Y-axis). To increase readability of Figure 2, the specific time points were added on top of the graphs (line 122).

Point 3: Figure 3 shows data on the accumulation of SYTOX/PI-stained yeast cells only at a concentration of 100 µg/ml. But what about other concentrations in a comparative aspect?!

Response 3: We selected the concentration of 100 µg/mL, equaling 4x MIC, as it is one of the concentrations that allows to select the highest number of cells that become PI/SYTOX-positive over the selected time frame (Figure 5).

Point 4: Fundamentally, the question is extremely important: for what reason did the authors use the approach in which the AMP concentration remained constant throughout the entire observation period? In general, if we try to imagine a “plant-phytopathogen” system, then a situation is unlikely in which the concentration of AMP would remain constant at the site of fungus inoculation, even for up to 4 hours. For defensins as induced polypeptides, the likelihood of a local increase in concentration would be higher. The opposite situation can be assumed, for example, with intravenous administration of a drug containing defensin, when trying to treat a fungal infection - in this case, the concentration of the active substance decreased over time. How is it even possible to extrapolate the current results on S. cerevisiae to a real living system? It would be more correct, at least selectively, to compare the data obtained with a constant, decreasing and increasing concentration of the defensin applied.

Response 4: We thank the reviewer for this interesting comment. For now, our main goals were to provide proof-of-concept (i) that we can investigate the kinetics of cellular events upon HsAFP1 treatment and (ii) that we can investigate the mode of action of HsAFP1 by multiplexing fluorescent dyes. In a next step, it would be highly interesting to vary the peptide concentration during the experiment, thereby more closely mimicking the natural conditions, which is feasible via the automated reagent delivery on-chip. To clarify this better, we included following paragraph in the discussion section (lines 303-309). “Lastly, here, we kept the concentration of HsAFP1 constant throughout the experiment. However, in plant – phytopathogen systems, a local and transient increase in plant defensin concentration is most likely to occur whereas when treating a fungal infection with intravenous administration of a defensin-based drug, the concentration of the active substance will decrease over time. Therefore, in follow-up experiments, it would be very interesting to assess the effect of varying the peptide concentration over time via the automated reagent delivery on-chip.”

Point 5: Line 215-216 - a surprising result, in which the largest number of SYTOX-positive cells were detected at 25-100 µg/ml. But what about at 300 µg/ml? The cells were completely destroyed?!

Response 5: We believe there is a misunderstanding regarding this data. When analyzing the time between reactive oxygen species (ROS) induction and membrane permeabilization within a single cell, not all single cells can be retained in this analysis. More specifically, cells that were already DHE- or SYTOX-positive at the start of the measurements, i.e. at 60 minutes after the start of HsAFP1 treatment, cannot be included in the analysis. As we have no data regarding the exact time point when these cells became fluorescent, it is impossible to define the time between ROS induction and membrane permeabilization. The number of cells indicated in lines 215-216 merely indicates the number of single cells that is included in this analysis. Upon treatment with 300 µg/mL HsAFP1, only 18% of the single cells were still SYTOX-negative at the start of image analysis (60 min), explaining the limited number of cells, being 67, that could be retained in the analysis to determine time between cellular events. We clarified this in the manuscript (lines 214-215).

Point 6: For all experiments, there is no data on what was the average number of stained cells in the control, that is, without the addition of the defensin? It is obvious that in any microbial culture there is a natural process of destruction (autolysis).

Response 6: Indeed, when optimizing the continuous microfluidic platform we evaluated the average number of stained cells in the untreated control yeast culture (i.e. without the addition of the defensin (Breukers et al. 2021). Here, we were able to demonstrated that confining single yeast cells in microwells for 4 hours did not affect cell viability. We included this explanation in more detail in the Results section (line 77).

Point 7: Is it possible to test these methods on antifungal AMPs, representatives of other structural families, for example, thionins and cyclotides? First of all, in terms of the rate of permeabilization of the fungal cell membrane.

Response 7: As thionins and cyclotides are known for their rapid killing kinetics (Thevissen et al. 1996, Ganesan et al. 2021), it would be possible to analyze their mechanism of action only if the delayed imaging start time (starting from 60 min after the start of treatment – due to cell docking and washing steps) can be reduced. This can be achieved by starting treatment after cell seeding and washing, although further optimization is required to accomplish this. Alternatively, since cell sedimentation via gravitation is time consuming, this process could be accelerated by employing strategies such as centrifugation (Huang et al. 2015) or by applying negative pressure (Swennenhuis et al. 2015).

Point 8: To what extent, from the point of view of the authors, is it possible to compare the membrane-active mechanism of action of AMPs with such well-known antimycotics as amphotericin B and azoles? How is HsAFP1 doing in terms of its cytotoxicity to mammalian cells?

Response 8: Indeed, it should be possible to monitor the response of single yeast cells upon treatment with amphotericin B and azoles using our microfluidic platform. Since both amphotericin B and azoles induce the production of reactive oxygen species (ROS) (François et al. 2006, Vriens et al. 2017) the kinetics of ROS induction (and potentially also of other cellular events) can be compared between treatment with well-known antimycotics, such as amphotericin B and azols, and plant defensins, such as HsAFP1.

Regarding the cytotoxicity of HsAFP1: Cools and colleagues demonstrated that HsAFP1 is not cytotoxic to human liver tumor cells (HepG2) (Cools et al. 2017b). We included this information in the Introduction section (lines 53-55).

References:

Breukers, J., Horta, S., Struyfs, C., Spasic, D., Feys, H.B., Geukens, N., Thevissen, K., Cammue, B.P.A., Vanhoorelbeke, K., Lammertyn, J. (2021) Tuning the surface interactions between single cells and an OSTE+ microwell array for enhanced single cell manipulation. ACS Applied Materials & Interfaces, 13(2), 2316-2326.

Cools, T.L., Vriens, K., Struyfs, C., Verbandt, S., Ramada, M.H.S., Brand, G.D., Bloch, C., Koch, B., Traven, A., Drijfhout, J.W., Demuyser, L., Kucharíková, S., Van Dijck, P., Spasic, D., Lammertyn, J., Cammue, B.P.A., Thevissen, K. (2017a) The antifungal plant defensin HsAFP1 is a phosphatidic acid-interacting peptide inducing membrane permeabilization. Frontier in Microbiology, 2017, 8, 2295.

Cools, T.L., Struyfs, C., Drijfhout, J.W., Kucharíková, S., Lobo Romero, C., Van Dijck, P., Ramada, M.H.S., Bloch, C., Cammue, B.P.A. and Thevissen, K. (2017b) A linear 19-mer plant defensin-derived peptide acts synergistically with caspofungin against Candida albicans biofilms. Frontiers in Microbiology, 8:2051

François, I.E.J.A., Cammue, B.P.A., Borgers, M., Ausma, J., Dispersyn, G.D., Thevissen, K. (2006) Azoles: mode of antifungal action and resistance development. effect of miconazole on endogenous reactive oxygen species production in Candida albicans. Anti-Infective Agents in Medicinal Chemistry, 5(1), 3-13.

Ganesan, R., Dughbaj, M.A., Ramirez, L., Beringer, S., Aboye, T.L., Shekhtman, A., Beringer, P.M., Camarero, J.A. (2021) Engineered cyclotides with potent broad in vitro and in vivo antimicrobial activity. Chemistry, 27(49), 12702-12708.

Huang L, Chen Y, Chen Y, Wu H (2015) Centrifugation-assisted single-cell trapping in a truncated cone-shaped microwell array chip for the real-time observation of cellular apoptosis. Analytical Chemistry, 87(24), 12169-12176.

Struyfs, C., Cools, T.L., De Cremer, K., Sampaio-Marques, B., Ludovico, P., Wasko, B.M., Kaeberlein, M., Cammue, B.P.A., Thevissen, K. (2020) The antifungal plant defensin HsAFP1 induces autophagy, vacuolar dysfunction and cell cycle impairment in yeast. Biochimica et Biophysica Acta – Biomembranes, 1862:183255.

Swennenhuis, J.F., Tibbe, A.G., Stevens, M., Katika, M.R., van Dalum, J., Tong, H.D., van Rijn, C.J., Terstappen, L.W. (2014) Self-seeding microwell chip for the isolation and characterization of single cells. Lab on a Chip, 15(14), 3039-3046.

Thevissen, K., Ghazi, A., De Samblanx, G.W., Brownlee, C., Osborn, R.W., Broekaert, W.F. (1996). Fungal membrane responses induced by plant defensins and thionins. J. Biol. Chem. 271(25), 15018-15025. DOI: 10.1074/jbc.271.25.15018.

Vriens, K., Tewari Kumar, P., Struyfs, C., Cools, T., Spincemaille, P., Kokalj, T., Sampaio-Marques, B., Ludovico, P., Lammertyn, J., Cammue, B., Thevissen, K. (2017) Increasing the fungicidal action of amphotericin B by inhibiting the nitric oxide-dependent tolerance pathway. Oxidative Medicine and Cellular Longevity, Art.No. 4064628.